# Enhanced Out-of-Stock Detection in Retail Shelf Images Based on Deep Learning

**DOI:** 10.3390/s24020693

**Published:** 2024-01-22

**Authors:** Franko Šikić, Zoran Kalafatić, Marko Subašić, Sven Lončarić

**Affiliations:** Image Processing Laboratory, Faculty of Electrical Engineering and Computing, University of Zagreb, 10000 Zagreb, Croatia; zoran.kalafatic@fer.hr (Z.K.); marko.subasic@fer.hr (M.S.); sven.loncaric@fer.hr (S.L.)

**Keywords:** deep learning, image analysis, image processing, out-of-stock detection

## Abstract

The term out-of-stock (OOS) describes a problem that occurs when shoppers come to a store and the product they are seeking is not present on its designated shelf. Missing products generate huge sales losses and may lead to a declining reputation or the loss of loyal customers. In this paper, we propose a novel deep-learning (DL)-based OOS-detection method that utilizes a two-stage training process and a post-processing technique designed for the removal of inaccurate detections. To develop the method, we utilized an OOS detection dataset that contains a commonly used fully empty OOS class and a novel class that represents the frontal OOS. We present a new image augmentation procedure in which some existing OOS instances are enlarged by duplicating and mirroring themselves over nearby products. An object-detection model is first pre-trained using only augmented shelf images and, then, fine-tuned on the original data. During the inference, the detected OOS instances are post-processed based on their aspect ratio. In particular, the detected instances are discarded if their aspect ratio is higher than the maximum or lower than the minimum instance aspect ratio found in the dataset. The experimental results showed that the proposed method outperforms the existing DL-based OOS-detection methods and detects fully empty and frontal OOS instances with 86.3% and 83.7% of the average precision, respectively.

## 1. Introduction

Out-of-stock (OOS) is a term that refers to a situation in which customers at a retail outlet arrive at the shelf and the specific product they are seeking is not available on its designated shelf [1]. Figure 1 shows an example of an image from a supermarket where multiple OOS situations are present on shelves. The OOS problem has been a matter of research for over 50 years, and supermarket shelves are still empty [2]. Corsten and Gruen [3] examined the extent, causes, and efforts to address the OOS problem. The study showed that the average worldwide OOS rate was 8.3%, with Europe having the highest rate of 8.6%. When customers encountered OOS, the average worldwide reactions were the following:There were 31% that bought the item at another store;There were 26% that bought an item of another brand;There were 19% that bought a different item of the same brand;There were 15% that delayed purchase;There were 9% that did not buy the item.
In particular, product categories with high brand loyalty, such as feminine hygiene products, diapers, and toothpaste, were most often bought at another store, while categories to which customers are less loyal, e.g., salted snacks and paper towels, were most often substituted with a different brand. A repetitive occurrence of OOS may result in a declining reputation or decreasing loyalty to both the store and the brand. Missing products generate huge losses in retailer sales with an estimated worldwide average of 3.9%. Furthermore, late shelf replenishment with products from storage was detected as one of the main causes of OOS, with a worldwide average share of 25%, while in Europe, the share was a massive 38%. Hence, an efficient and accurate OOS detection system could increase sales by 1%.

Shelf replenishment with products from storage can be late even up to 17 h, whereas OOS caused by other reasons sometimes remains unresolved for several days [4]. Currently, store clerks perform visual inspections of store shelves to identify possible OOS. Such a process is labor-intensive and depends on the frequency of physical visits to each shelf. Therefore, the implementation of a high-performing OOS detection system would not only increase the volume of sales, but also relieve the employees of the visual inspection task and ensure the possibility of employing them in other store-managing tasks.

Numerous approaches and different data can be used to address the problem of OOS detection. The simplest solution is to analyze the sales data [5] and track the quantity of unique products sold. If many items of a unique product have been sold since opening or, even more precisely, if the ratio of sold items to displayed items of a product is large, store employees should be notified to replenish the appropriate shelf. Alternatively, different sensors can be installed and utilized to detect changes on store shelves and possible OOSs, such as radio frequency identification (RFID) technology [4], infrared (IR) sensors [6], or depth sensors [7]. Furthermore, computer vision methods can be utilized to detect OOS in images of store shelves. In recent years, deep learning (DL) networks, which process input data using a large sequence of different layers (e.g., convolutional, pooling, and fully connected), have been employed in various tasks, including OOS detection in images. DL-based methods provide the possibility of OOS detection in images using two opposite strategies, either by detecting products on store shelves and deducing the locations of OOSs [8,9,10] or by direct detection of OOS instances present in an image [11,12]. Also, OOSs may be detected using on-shelf availability (OSA) estimation [13,14], a task in which the goal is to estimate the amount of products on store shelves, as a minimal OSA estimation for a certain shelf suggests that an OOS is present on the shelf.

A major problem in OOS detection in images is the data. In particular, only one OOS detection image dataset [15] has been made publicly available so far. However, OOS instances from the mentioned dataset were marked using only one point, thus not capturing the height and width of the objects, which is a drawback that makes the dataset useless for training DL object-detection models. Therefore, any researcher who plans to study this problem and perform certain DL experiments must collect and annotate his/her own set of images. Next, there are many questions regarding data collection and annotation processes. When and how do we collect the images? How do we annotate the height of an empty area on the top shelf? How do we label continuous empty space on a shelf? How dow we deal with small gaps on a shelf? These questions had not been answered until a recent study [12] became the first to reflect on these issues. An explicit definition of the data collection and annotation guidelines is crucial for the acquisition of a dataset with all four data quality dimensions mentioned by Batini et al. [16]: accuracy, completeness, consistency, and timeliness.

In this work, we propose a novel DL-based method for direct OOS detection in images of supermarket shelves. To develop the method, we utilized an OOS detection dataset that contains two OOS classes: fully empty OOS locations and frontal OOS locations (detailed description in Section 3). The proposed method is divided into a training part, which consists of three processes, and an inference part, which is performed through two steps. We present a new image-augmentation procedure suitable for OOS detection. In the training part, the proposed augmentation procedure is first applied to obtain augmented shelf images. Following the shelf images’ generation process, these generated images are used to pre-train an object-detection model to detect fully empty and frontal OOS instances directly. Finally, the pre-trained model is fine-tuned using the original images. The pre-training step and the fine-tuning step represent the first stage and the second stage of the multi-stage model training process, respectively. During the inference, an image is first passed through the OOS detection model. Later, the obtained results are post-processed as follows. For each detected OOS instance, the bounding box aspect ratio is calculated and compared with the aspect ratio of the instances present in the dataset. The detected OOS is discarded if its aspect ratio is either higher than the maximum or lower than the minimum OOS aspect ratio found in the dataset. Otherwise, the detected OOS instance is considered a valid detection.

The main contributions and novelty of this paper are as follows:We are the first to present an effective multi-stage training setup for object-detection networks that detect OOS directly. We show that the proposed two-stage training procedure improves the performance of the baseline models, i.e., of the models trained on the original images without pre-training on the augmented images.We are the first to introduce a post-processing technique designed for the removal of inaccurate OOS detections. We show that the proposed post-processing technique further improves the results obtained by the model developed using the two-stage training procedure.We are the first to use a frontal OOS class. We show that using frontal OOS instances not only enables the model to detect an additional OOS class, but also improves the results of the fully empty OOS class.

Following the Introduction, existing methods for OOS detection are thoroughly described in Section 2. A description of our in-house OOS detection dataset is provided in Section 3, followed by a detailed explanation of the proposed method in Section 4. The setup used in the conducted experiments is described in Section 5. Section 6 presents a quantitative and visual evaluation of the proposed method and a discussion of the obtained results. Finally, Section 7 gives the concluding remarks of the paper.

## 2. Related Work

OOS detection is a problem that can be solved using various approaches. As one of the first solutions to this problem, Papakiriakopoulos et al. [5] proposed a rule-based decision support system for automatic OOS detection based on sales and other data. Later, several studies proposed a sensor-based solution for the detection of OOSs on store shelves. Bertolini et al. [4] used the RFID technology to track the number of products on supermarket shelves. Frontoni et al. [6] built an embedded wireless IR sensor network for real-time OOS detection. Milella et al. [7] presented a solution for OSA estimation using 3D data provided by a depth sensor. Moorthy et al. [17] mentioned another possible solution, where a weight sensor is integrated into a shelf and a change is detected whenever a product is taken from the shelf. Over the last decade, computer vision applications have attracted increasing attention. Therefore, multiple image processing (IP) and DL solutions for OOS detection have been presented. In comparison to the sensor-based approaches, computer vision-based methods are less expensive to integrate into an OOS detection system deployed in a supermarket, have less scalability issues, and are able to analyze the status of more products at once [18,19].

Rosado et al. [20] proposed a framework for OOS detection in panoramas of retail shelves using IP and machine learning. Following the image stitching process based on the research of Goncalves et al. [21], OOS detection was performed through several processing steps. First, the FAST corner detection algorithm [22] was applied to a panoramic image to detect keypoints, and the locations of the aisle edges were deduced based on the vicinity of the keypoints. The detected keypoints were then utilized to construct a binary mask of the OOS candidates inside the localized aisle area. The mask was later divided into rectangles and binarized according to the density of keypoints inside the rectangle. Vertical separation of the OOS candidates present in the mask was followed by Hue-Chroma-Luminance color space-based filtering of the candidates. Each remaining candidate was described using a total of 152 image features from three different feature categories: geometry, texture, and color. Finally, a two-class support vector machine (SVM) [23] performed the classification of candidates using the aforementioned image features.

OOS can be localized by detecting products present on the shelf and analyzing areas where products have not been detected. Šećerović and Papić [8] built such a system for the detection of missing products in commercial refrigerators using convolutional neural networks (CNNs). First, Faster R-CNN [24] and SSD [25] object detectors were trained to localize products on shelves. The OOS positions were then deduced according to the locations of the goods as follows. The K-means clustering algorithm [26] was used to group the products detected on the same shelf using only the y coordinate of the detections. The average silhouette method [27] was applied to determine the correct number of groups, i.e., shelves on an image. Finally, a significant distance between products detected on the same shelf implied the presence of an OOS problem. Chen et al. [9] developed a method for OOS detection based on product detection and various image analysis techniques. First, an object detection CNN was used to detect products in images of store shelves. In particular, the Faster R-CNN model with the Inception_v2 [28] backbone was trained for this purpose. The positions of the detected products were then used to determine the locations of the unknown areas on the shelves. The unknown areas were finally analyzed using three separate approaches: analysis of the amount of edge information extracted using the Canny edge operator [29], classification of texture features using an SVM, and comparison with the pre-computed color histograms of typical OOS areas. Achakir et al. [10] detected OOSs by employing Faster R-CNN and MiDaS [30] models for product detection and depth estimation in a shelf image, respectively. The results obtained with the Faster R-CNN were refined using the ASIFT [31] descriptor to increase the product detection accuracy. Finally, the presence of an OOS in the image was deduced based on the mean estimated depth of areas where the products were not detected.

In addition to being employed in product detection-based OOS approaches, object-detection models can also be trained to detect OOS directly. Yilmazer and Birant [11] used semi-supervised DL to localize products and OOSs in grocery stores. First, three different one-stage object-detection models were trained to detect three product classes and two OOS classes (empty shelf and almost empty shelf) using the labeled data. In particular, RetinaNet [32], YOLOv3 [33] and YOLOv4 [34] models were selected for this purpose. Following the training process, the best-performing model was used to label the unlabeled data, and the predictions of the model were considered as pseudo-labels. Finally, the best-performing model was re-trained using both labeled and pseudo-labeled data to obtain the final model. Jha et al. [12] trained several EfficientDet [35] models and the complete family of YOLOv5 [36] models in order to detect OOS in shelf images. The influence of the training set size on the result was examined, and the obtained results showed an almost logarithmic increase in performance with a linear increase in the training set size.

In recent years, CNNs have also been employed in various tasks related to OOS detection. Higa and Iwamoto [13] published a method for robust OSA estimation using IP and a six-channel input CNN. First, background subtraction was performed to detect changes in the shelf image. Next, the Hungarian method [37] was used to determine the correspondence of foregrounds between consecutive images, and moving foregrounds were discarded. For each detected change, a six-channel image was constructed from the cropped region of change before and after the change was detected, and fed into a CNN that classified a shelf change. The classification results were used to update the shelf condition, which was represented as a binary image. Finally, the OSA metric was calculated as the ratio of the shelf area containing the products to the entire shelf area. The extended version of the study [14] proposed an additional functionality for generating a heatmap that shows the accumulation of detected changes on store shelves. Retailers can use such information to plan the most profitable product placement strategy.

Allegra et al. [15] proposed a CNN based on U-Net [38] that was utilized to predict attention maps useful for localizing the OOS present in an image. The authors also published a dataset that has remained the only publicly available OOS detection image dataset up to now. The dataset was built by re-annotating images from the Ego-Cart [39] dataset. In particular, each image was labeled with points that represent the central point of the OOS instances present in the image. Santra et al. [18] used graph-based modeling of superpixels to automatically segment empty areas on supermarket shelves. First, all the images were over-segmented into superpixels, i.e., regions that consist of a group of pixels, using the simple linear iterative clustering (SLIC) [40] algorithm. A graph was then constructed for every shelf image, with the superpixels as nodes and the edges for each pair of neighboring superpixels. A graph convolutional network [41] and Siamese network architecture [42] were utilized to obtain a unary feature embedding for each node and a pairwise feature embedding for each edge, respectively. Finally, a structural SVM [43], a generalization of SVMs that can be used to address a large range of structured output prediction tasks [44], classified nodes into a gap or non-gap class using the following structured data as input: adjacency list of the superpixel graph, unary feature embedding of the nodes, and pairwise feature embedding of the edges.

Although in [9] the authors claim that direct OOS detection using a CNN cannot provide satisfactory results due to the inconsistency of the size, illumination, and background of OOSs, the methods proposed in [11,12] show that not only satisfactory but also very accurate results may be achieved using such an approach. However, none of the proposed methods for direct OOS detection present any post-processing technique or custom training setup that manages to improve the results obtained by utilizing a basic CNN training procedure (in [11] semi-supervised learning did not improve the result). The addition of these techniques may be crucial for further improving the results of the OOS detection CNNs. Furthermore, only one of the existing OOS-detection methods uses an additional OOS class along with a fully empty class. In particular, in [11] the authors used an ‘almost empty’ OOS class. However, they did not describe what ‘almost empty’ means, how the class was annotated, or what impact it had on the results of the fully empty class. A clear description of the additional OOS class is mandatory for a well-performed data annotation process.

## 3. Dataset

As in any other research area, a good dataset is of utmost importance in DL. As previously mentioned, the only publicly available OOS detection image dataset was published in [15] and contains images in which OOS instances were labeled using one point. However, using only one point in an object detection task is not sufficient to localize the instances precisely, as information on the width and height of the instances is missing. Although it is sometimes possible to convert datasets from one task to another, e.g., converting instance segmentation maps into instance bounding boxes, in this case, the dataset cannot be appropriately adjusted as it is impossible to convert from instance points to instance bounding boxes. Therefore, we collected and manually labeled a set of 511 RGB shelf images from four major retailers located in Croatia. The images capture the full shelf height from top to bottom and were taken in aisles that contain various products, such as beverages, coffee, and pâtés. The resolution of the acquired images varies from 1800 to 4600 pixels in both the height and width.

We labeled the data using the same annotation guidelines as those presented in [12]. These guidelines focus only on completely empty shelf locations, i.e., locations empty from front to back and top to bottom of the shelf. The empty location was visualized as a three-dimensional (3D) box, and the front face of the 3D box was labeled. In the scenario where multiple neighboring products were taken from the shelf, the continuous empty location was labeled as a single empty location instance. Sometimes, small empty locations are present on the shelf due to human interference and therefore, only locations with at least half the size of the neighboring products were labeled. Furthermore, we extended these annotations with an additional semi-empty location class. In particular, we also visualized a 3D box for the location where multiple products were taken from the front of the shelf, but some products were still present at the back of the shelf, and labeled the bottom face of the 3D box if visible. We refer to the aforementioned completely empty locations as the normal class and additional semi-empty locations as the front class. Monitoring the occurrence of the front class may be beneficial to retailers as its presence can be considered an early warning for a potential new normal class instance. Figure 2 displays examples of OOS instances present in the dataset. In certain scenarios, the backside of the shelf is not solid, and the background scene is visible through the shelf, sometimes even including recognizable products. Although OOSs are generally characterized by a lack of keypoints and homogeneous texture, an accurate OOS detection model should be able to detect even these challenging OOS instances.

Table 1 shows the distribution of the images and OOS instances for different store sections. Each store section is represented by approximately 170 images which contain from 322 to 446 OOS instances. Whereas images from the beverage and coffee sections display shelves with twice as many normal class instances as front instances, images from the pâté section have an even ratio of instances of both classes. The average count of OOS instances in an image ranges from 1.8 to 2.8 for different store sections, making a total average of 2.4 OOSs per image across the entire dataset. In particular, Figure 3 illustrates the distribution of OOS instances per image. The number of OOS instances varies from zero to 15, 10, and 11 in the beverage, coffee, and pâté sections, respectively. In general, the number of images decreases exponentially with a linear increase in the number of OOS instances in an image.

## 4. Method

The proposed OOS-detection method is illustrated in Figure 4. We divide our method into two main parts: a training section and an inference section. First, a new augmentation procedure suitable for OOS detection is applied to the original shelf images that contain OOSs to produce augmented shelf images as follows. A randomly selected OOS instance from an original image is duplicated and mirrored horizontally, either to the left or right, thus enlarging itself over nearby products. The proposed OOS mirroring augmentation technique is suitable for OOS detection in images for two reasons. First, the mirroring technique realistically demonstrates a real-world process in which various shoppers take products from the same shelf throughout the working day. Second, usual data augmentation techniques, such as image translation and scale shift, change the absolute location of OOS instances in images but, unlike the proposed mirroring augmentation, do not alter the shape of the instances. An object-detection model trained using images produced by the proposed mirroring technique may benefit from extended instances to achieve a better localization performance.

The direction of mirroring *D* is selected based on the location of the center point *C* of the selected OOS instance *O* in the shelf image *I*, as expressed in (1). If *C* is located in the left quarter of *I* (i.e., the horizontal coordinate of the center point *C_x_* is less than a quarter of the width of the image *I_width_*), then *O* should be mirrored onto the right side (*RS*). Similarly, if *C* is located in the right quarter of *I* (i.e., *C_x_* is higher than three-quarters of *I_width_*), then *O* should be mirrored onto the left side (*LS*). Otherwise, *C* falls within the middle part of *I* and *D* is chosen randomly.
(1)$$\begin{array}{c} D=\left\{\begin{array}{cc} RS, & \mathrm{if}\;C_{x}<\frac{1}{4}\cdot I_{\mathrm{width}}\\ LS, & \mathrm{if}\;C_{x}>\frac{3}{4}\cdot I_{\mathrm{width}}\\ random(LS,RS), & \mathrm{otherwise} \end{array}\right. \end{array}$$

For OOS instances located near the left and right edges of an image, the heuristic given in (1) selects *D* towards the center of the image. This is done because mirroring a near-edge instance towards the edge would often make only a minor alteration to the instance by expanding it to the edge in the first iteration, while if the same instance is chosen again in the second iteration, then mirroring towards the edge would not alter the instance at all. Furthermore, the thresholds are arbitrarily set to a quarter away from the left and right image edges and may be replaced with other values, such as a third or fifth away from the vertical edges. However, note that using smaller thresholds (i.e., closer to the vertical edges) may result in a significant number of edge-touching instances in the first iteration, which, as explained earlier, may not be affected by the OOS mirroring technique in the second iteration. On the other hand, using large thresholds (i.e., closer to the image center) may cause one mirroring direction to be more favorable if the horizontal distribution of instances is shifted towards one of the vertical edges.

While mirroring the newly duplicated OOS, we avoid making overlaps with other OOS instances present in the image. In such a scenario, we repeatedly reduce the width of the newly duplicated OOS by *T* times until there is no overlap with other OOS instances. In our research, we set the value of *T* to 0.75. The proposed augmentation technique can be repeated through *N* iterations to produce *N* new images from a single original image. In particular, in the first iteration, the augmentation technique is applied to the original shelf image *I* to produce the first augmented image *I*_1_. During the next iterations, an augmented image *I_N−1_* produced in the previous iteration is used as the input to which the augmentation technique is applied to produce a new augmented image *I_N_*. In our research, we repeat the process for two iterations and therefore obtain 786 new shelf images using 393 original images that contain OOS instances. The pseudocode for applying the proposed OOS mirroring procedure to a set of images and the corresponding annotations is displayed in Algorithm 1, whereas Figure 5 shows an example of the original shelf image and new augmented shelf images produced by the aforementioned procedure. In the pseudocode, each function (e.g., *randomly_select_OOS* and *find_mirroring_direction*) performs a specific task, as implied by its name, using the passed arguments.
**Algorithm 1 **OOS mirroring procedure1:**procedure** Augment_Dataset(*images, annotations*)2:      *T*, *N* = 0.75, 2                           ▹ Width shortening and iterations count constants3:      **for each** *I* ∈ *images* **do**4:          *image_annotations* = *select_image_data*(*I*, *annotations*)5:          **for** *iteration* = 1 **to** *N* **do**6:                *O* = *randomly_select_OOS*(*image_annotations*)7:                *D* = *find_mirroring_direction*(*O*, *I*)8:                *EC* = *find_enlargement_coordinates*(*O*, *D*)             ▹ *O* is mirrored onto *EC* area9:                **while** *intersection_exists*(*EC*, *image_annotations*) **do**10:                    *EC* = *reduce_width*(*EC*, *T*, *D*)11:                **end while**12:                *new_image* = *mirror_OOS*(*I*, *O*, *EC*)13:                *new_image_data* = *adjust_OOS_coordinates*(*image_annotations*, *O*, *EC*) ▹ *O*_*new*_ = *O* + *EC*14:                *save*(*new_image*, *new_image_data*)15:                *I*, *image_data* = *new_image*, *new_image_data*16:          **end for**17:      **end for**18:**end procedure**

Next, the generated augmented images are used to pre-train an object-detection model for the OOS detection task. In particular, we chose the YOLOv5, YOLOv7 [45], and EfficientDet models for this purpose, but the aforementioned model can be any other existing (or future-developed) object-detection model. Following the pre-training step, the model is finally fine-tuned using the original dataset. The two-stage training procedure results in an OOS detection model that can be deployed in a supermarket system by using only the inference part of the method. Since the chosen object-detection models are fully convolutional, the deployed model can accept RGB input images of an arbitrary resolution. However, the best performance is expected when the input image resolution is the same or very similar to the one used during the training process.

During inference, a previously unseen shelf image is passed through the model to obtain the initial OOS detection results. Each detected OOS instance contains information about the position, height, width, class, and confidence score of the OOS present in the image. Later, the detected OOS instances with extreme aspect ratios are discarded as follows. Let *a* be a bounding box aspect ratio defined as:(2)$$a=\frac{OOS_{\mathrm{height}}}{OOS_{\mathrm{width}}}.$$
We performed exploratory data analysis of our data and calculated *a* for each OOS instance present in the dataset. Table 2 showcases the obtained intervals of *a* for each class and store section, as well as the intervals for the entire dataset. If *a* of the OOS instance detected by a model trained using a particular store section data falls within the calculated range of the used store section and the predicted class of the detected instance, the detection is considered valid; otherwise, it is discarded. For example, if a model trained using the beverage store section data detects an OOS instance of class normal, then *a* of the detected instance has to fall within the range from 0.27 to 4.53 to be considered a valid detection. The end of Section 5 provides additional implementation details regarding the post-processing of the detected OOS instances.

## 5. Experimental Setup

To validate the proposed method, we trained the YOLOv5, YOLOv7, and EfficientDet object-detection models. In particular, a small version of the YOLOv5 model from the sixth release of the homonymous GitHub repository, YOLOv5s6, and similar-sized versions of the YOLOv7 and EfficientDet, YOLOv7-tiny and EfficientDet-D3, were utilized for this purpose. Small versions of these models were selected as they are suitable for running on mobile phones and other resource-limited devices. The models were trained using the following setup. Weights from the models pre-trained on the COCO [46] dataset were used as the starting point in each experiment. The models were optimized using the stochastic gradient descent with the Nesterov momentum [47] of 0.937. The learning rate lr(e) was decreased linearly, as defined by the following formula [36]:(3)$$lr\left(e\right)=lr_{\mathrm{init}}\cdot \left[\left(1-\frac{e}{E}\right)\cdot \left(1-lrf\right)+lrf\right],$$
where *lr_init_* represents the initial learning rate, *e* is the epoch for which the learning rate is calculated, *E* represents the total number of training epochs given at the start of training, and *lrf* is the learning rate factor that controls the amount of decrease. In our experiments, the models utilized 0.01 for both *lr_init_* and *lrf*, whereas 1000 was used as *E*. Furthermore, the models were trained using a set of augmentation techniques including translation, scale shift, horizontal flip, HSV color space channels modulation, and mosaic [34]. The aforementioned training hyper-parameters and setup are the default values and setup from the GitHub repositories of the trained models. The input images were scaled to a 1280 × 1280 resolution using letterbox scaling, i.e., the aspect ratio of the original images was preserved. The models were trained on a single NVIDIA GeForce RTX 2080 Ti graphics card using a batch size of eight.

The dataset was randomly shuffled and split into five equal-sized cross-validation folds. Also, we decided to leave out 15% of the training subsets to form the validation subsets, which were utilized for the early stopping of the model to avoid overfitting. The produced training, validation, and test subsets that contain the original images were used to fine-tune the models. Before fine-tuning, the models were pre-trained on the augmented images using additional training subsets formed as follows. The training subsets used for pre-training were constructed only with shelf images obtained by applying the OOS mirroring procedure to the original images present in the training subsets used for fine-tuning the model. In particular, if a training subset used for fine-tuning contains an original image *I*, the training subset used for pre-training should contain images *I*_1_ and *I*_2_, which were produced by applying the proposed augmentation method to the original image *I* through two iterations. It is of utter importance to follow this setup to prevent data leakage.

We used two different pre-training strategies: pre-training of a model for a fixed number of epochs and pre-training of a model in which early stopping was performed based on the results of the validation subset. In the latter strategy, the validation subset used during pre-training was the same as the one later used for fine-tuning the model. To prevent data leakage during the validation of the post-processing technique, instead of using the thresholds presented in Table 2, we recalculated the thresholds for each test subset based only on the corresponding training subset data and used them to post-process the initial results of the test subset for which they were calculated.

## 6. Results and Discussion

We evaluated the effectiveness of the proposed method using a commonly used class-wise object detection metric of average precision (AP) [48] and its multi-class counterpart mean AP (mAP). Table 3 shows the results obtained with the proposed method and a comparison with methods that utilize two main deep learning-based OOS detection strategies: product detection-based OOS detection [9] and direct OOS detection [12]. Since product detection-based methods rely on analyzing unknown areas where products are recognized to the left and/or right of the unknown area, they are not able to detect the front OOS class, which is surrounded by products from behind (i.e., upper than the front instance in an image) as well. Therefore, the different methods presented in [9] were evaluated only on the normal OOS class. All product detection-based methods performed significantly worse than direct detection-based methods, achieving 20% to 25% lower AP on the normal OOS class. The proposed method (i.e., YOLOv5/YOLOv7/EfficientDet + F) achieved approximately 4% higher AP on the normal class than the best-performing existing method (i.e., Jha et al. [12] using YOLOv5/YOLOv7/EfficientDet and the normal class only). To evaluate the impact of using the front class, we additionally trained the proposed method without the front class and the best-performing existing method using both the normal and the proposed front classes. The obtained results show that the use of the front OOS class not only ensured the ability to detect additional OOS instances with high accuracy, but also affected the results of the normal class, which were increased by approximately 1%. Furthermore, the results show that the YOLOv5 models consistently outperformed the YOLOv7 and EfficientDet models in all experiments.

Table 4 displays the results obtained with the YOLOv5, YOLOv7, and EfficientDet models using the proposed two-stage training procedure. Each model was trained and validated using individual store section data as well as using the entire dataset. The baseline models were trained for direct OOS detection of both normal and front classes using only the original images (i.e., without pre-training on the augmented images), following the method presented in [12]. In addition to pre-training the models using the early stopping-based strategy, several pre-training strategies that use a fixed number of epochs were utilized. In particular, models trained with beverage, coffee, and pâté section data were pre-trained for 30, 50, and 100 epochs, while the model trained with all of the data was additionally pre-trained for 10 epochs as using three times more data and the same batch size as the models trained with only a subset of the data enables the model to converge faster. Models pre-trained using the early stopping-based strategy generally trained for approximately 140 to 180 epochs.

First, we discuss the influence of different pre-training strategies on the results of the YOLOv5 models. For the beverage section, the best result was achieved using pre-training for 100 epochs, whereas models trained using images from the coffee and pâté sections performed the best when they were pre-trained for 50 epochs. However, when training with the complete dataset, the best-performing model was obtained using only 10 pre-training epochs. The optimal pre-training strategy ensured an increase in the result which ranged from 1.8% to 2.3% for different store sections, and an improvement of 1.6% for the entire dataset. Next, we discuss how the two-stage training procedure affects the performance of YOLOv7 models. For the beverage and pâté sections, the best results were achieved using pre-training for 50 epochs, whereas models trained using images from the coffee section and the complete dataset performed the best when pre-trained for 30 epochs. The best-performing pre-training strategy provided an improvement in the result ranging from 1.8% to 5.3% for different store sections, and an increase of 2.0% for the entire dataset. Finally, we discuss what an impact the two-stage training process had on the results of EfficientDet models. For the beverage and coffee sections, the best results were obtained after pre-training the models for 50 epochs, whereas models trained using images from the pâté section and the complete dataset performed the best when pre-trained for 30 epochs. The optimal pre-training strategy helped increase the results by 2.0% to 3.0% for different store sections, and by 2.1% for the entire dataset. In general, each YOLOv5 model outperformed the corresponding YOLOv7 and EfficientDet models.

Furthermore, to demonstrate the advantages of pre-training the models using images generated by the proposed OOS mirroring technique, we replaced the proposed technique with several other data augmentation techniques and compared the results. In particular, we compared OOS mirroring with rotation, cutout [49], and contrast. In the rotation experiments, the input images were randomly rotated by an angle up to ±20°. Cutout was applied by masking out 10 randomly selected locations in an image, where the mask height and width were randomly chosen to be from 10% to 15% of the image height and width, respectively. In the contrast experiments, the contrast of the input images was adjusted using a contrast factor randomly selected from the interval [0.5, 2.0]. Table 5 presents the results of the two-stage training procedure using the aforementioned data augmentation techniques. The proposed OOS mirroring technique outperformed other data augmentation techniques by at least 1.4%, 1.7%, and 1.8% for the YOLOv5, YOLOv7, and EfficientDet models, respectively.

The ablation study in Figure 6 shows the results obtained after applying the proposed post-processing technique to the baseline models and the optimally pre-trained models for each store section (e.g., the YOLOv5 model pre-trained on 100 epochs and later fine-tuned using beverage section data). The results achieved without post-processing are taken from Table 4 and put next to the post-processed results for a clear comparison. Post-processing of the results obtained with the baseline models trained on either a particular store section data or all of the data provided an improvement in the result ranging from 1.2% to 1.9%, from 1.3% to 2.4%, and from 1.3% to 2.5% for the YOLOv5, YOLOv7, and EfficientDet models, respectively. When the post-processing technique was applied to the optimally pre-trained models, the results improved from 1.2% to 1.8%, from 1.3% to 2.2%, and from 1.3% to 2.1% for the YOLOv5, YOLOv7, and EfficientDet models, respectively. Again, each of the YOLOv5 models achieved a higher result than the corresponding YOLOv7 and EfficientDet models.

The ablation study shows that both the two-stage training strategy and post-processing technique can improve the results of the baseline models individually, but once they are combined, the results improve even further. To understand what an impact the proposed method can have on a retailer, let us consider a supermarket chain with an annual revenue of $10 billion. In Section 1 of our manuscript, we showed that OOSs caused by late shelf replenishment from storage cause a sales loss of 1%, i.e., approximately $100 million. An OOS detection system that works with approximately 80% accuracy helps to achieve timely replenishment and increase sales (by selling the replenished products) by $80 million. The incorporation of the proposed method into the aforementioned system ensures an increase of approximately 4% in the result, equivalent to an additional $4 million in revenue.

Additionally, since the proposed method is intended for use in real-world systems, the runtime of the method needs to be discussed. Running the method on a GPU lasts 17 ms, 7 ms, and 61 ms per image for the YOLOv5, YOLOv7, and EfficientDet models, respectively. On the other hand, running on a CPU lasts 414 ms, 264 ms, and 3627 ms per image for the YOLOv5, YOLOv7, and EfficientDet models, respectively. It is important to note that OOS methods do not need to produce real-time results. Customers sometimes take the product and read the declaration in front of the shelf, after which they may take the product or return it to the shelf. Therefore, it is sufficient to analyze store shelves every minute or less frequently. Even if there are 100 cameras around the store and only one central unit backstage that analyzes images sequentially, it takes YOLOv5 (i.e., the best-performing model) approximately 40 s to analyze the entire store on a CPU (and less than 2 s on a GPU), which is well below the necessary analysis time.

The top row of Figure 7 displays images that are successfully analyzed by the optimal OOS detection model that was developed using the whole dataset, i.e., by the YOLOv5 pre-trained for 10 epochs and with post-processing included. In Figure 7a, the model is able to accurately localize and classify all five OOS instances, including the one on the top of the shelf where a lot of products can be seen through the shelf. In Figure 7b, one OOS instance is correctly detected, while two detected ruptures between products are successfully discarded by the proposed post-processing technique.

However, the optimal model sometimes struggles to detect OOS instances accurately. In certain scenarios, the model does not detect empty areas on store shelves, although the products are missing, i.e., a false negative (FN) occurs. Also, in some images, the model detects false positives (FPs), i.e., OOS instances are either detected in locations where they do not exist or are not localized accurately enough, thus having insufficient intersection over union ratio with the ground truth to be considered true positives (TPs). The bottom row of Figure 7 shows two images that contain, alongside accurate OOS detections, some inaccurate detections. In Figure 7c, an FP occurs on the second shelf from the bottom in the location where OOS is not present due to the products hanging at the end of the shelf. In Figure 7d, a huge OOS instance at the top of the shelf is detected inaccurately, localizing just a part of the actual empty space on the shelf.

## 7. Conclusions

This paper presents a novel DL-based method for OOS detection in images of supermarket shelves that capture the full shelf height. A lack of products on store shelves may result in sales losses, customer dissatisfaction, and a declining reputation. Hence, an efficient OOS detection system is highly valuable to any retailer. Missing products can be detected by using sales data or sensor-based approaches. However, the latest solutions mostly rely on computer vision-based methods.

The proposed method ensured highly accurate detection of both normal and front OOS instances, outperforming the existing deep learning-based OOS-detection methods by at least 3.9% of the AP on the normal OOS class. The use of the front OOS class proved to be beneficial as it not only enabled the model to recognize an additional OOS class, but also improved the results of the normal OOS class by approximately 1% of the AP. The presented OOS mirroring procedure enables the easy generation of new semi-realistic shelf images. These images can be used to pre-train an object-detection model for the OOS detection task. The proposed two-stage training procedure helped the models achieve better generalization performance, increasing the baseline results by up to 5.3% of the mAP for different store sections. When selecting the best training procedure, several pre-training strategies must be validated to determine the optimal strategy for a given dataset. Furthermore, the proposed post-processing technique proved to be beneficial for the removal of FPs caused by visible ruptures between different products on the shelf, additionally increasing the results by up to 2.2% of the mAP for different store sections.

In future work, depth estimation should be incorporated into the existing solution to further improve the results. Having depth information would be beneficial for better scene understanding and reducing the number of FNs and FPs.

## Figures and Tables

**Figure 1 sensors-24-00693-f001:**
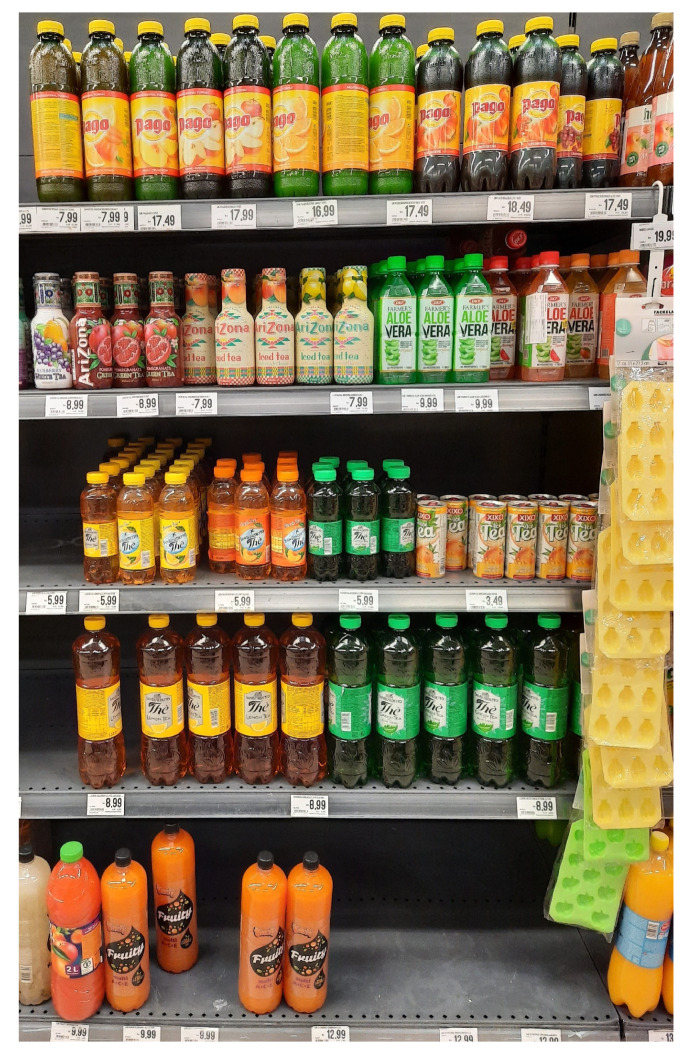
Example of an image from our dataset with several out-of-stock (OOS) locations on shelves.

**Figure 2 sensors-24-00693-f002:**
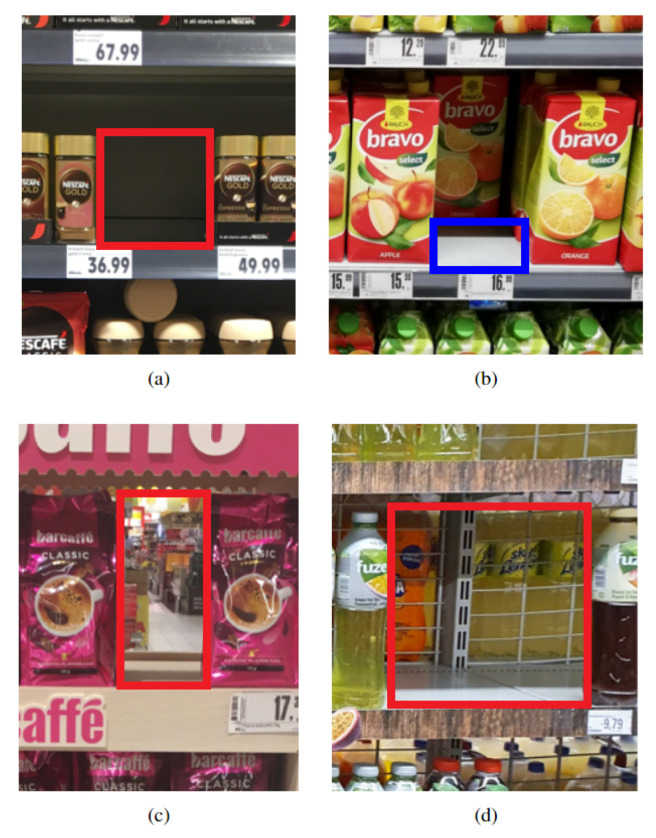
Examples of OOS instances. The red and blue rectangles mark the exact location of the normal and front class instances, respectively. The top row shows (**a**) normal and (**b**) front class instances, whereas the bottom row shows challenging normal class instances where (**c**) an unrecognizable background or (**d**) a recognizable product can be seen through the shelf.

**Figure 3 sensors-24-00693-f003:**
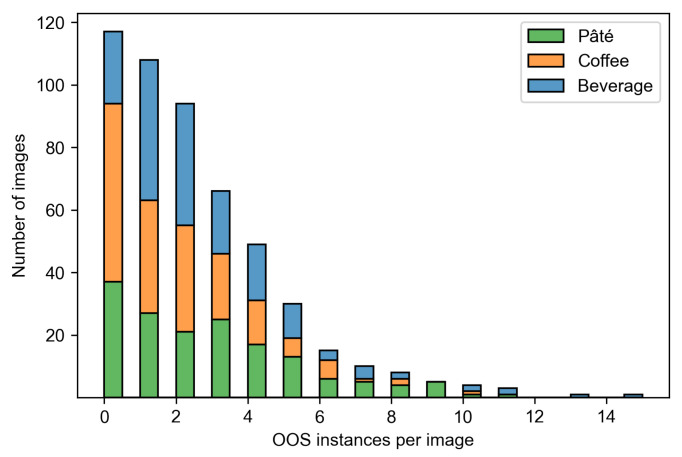
Histogram of OOS instances distribution per image. Each bar displays the cumulative count and the share of each store section.

**Figure 4 sensors-24-00693-f004:**
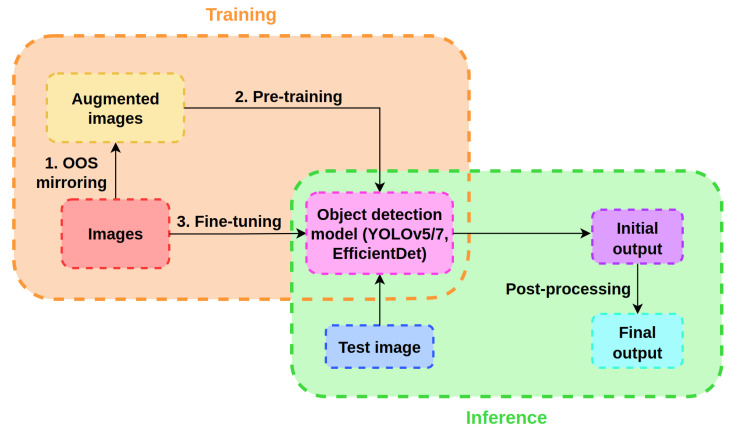
Scheme of the proposed OOS-detection method. The training part and the inference part of the method are marked with orange and green dashed rounded rectangles, respectively.

**Figure 5 sensors-24-00693-f005:**
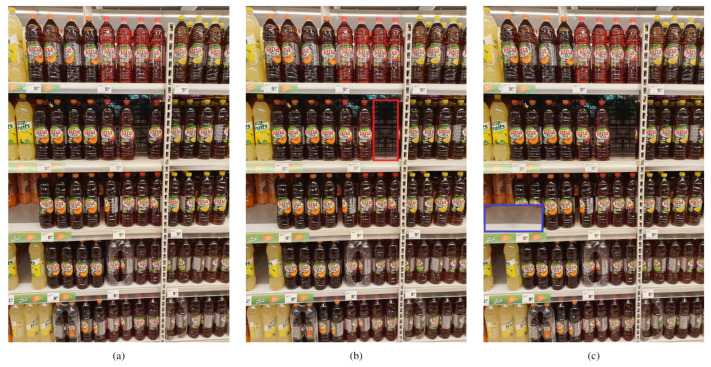
Example of the OOS mirroring procedure. In the first iteration, the proposed OOS mirroring technique is applied to (**a**) the original image to produce (**b**) the first augmented image. In the second iteration, the augmentation technique is applied to (**b**) to produce (**c**) the second augmented image. For each iteration, the newly extended OOS instance is marked with a color-coded rectangle, where red and blue colors represent normal and front classes, respectively.

**Figure 6 sensors-24-00693-f006:**
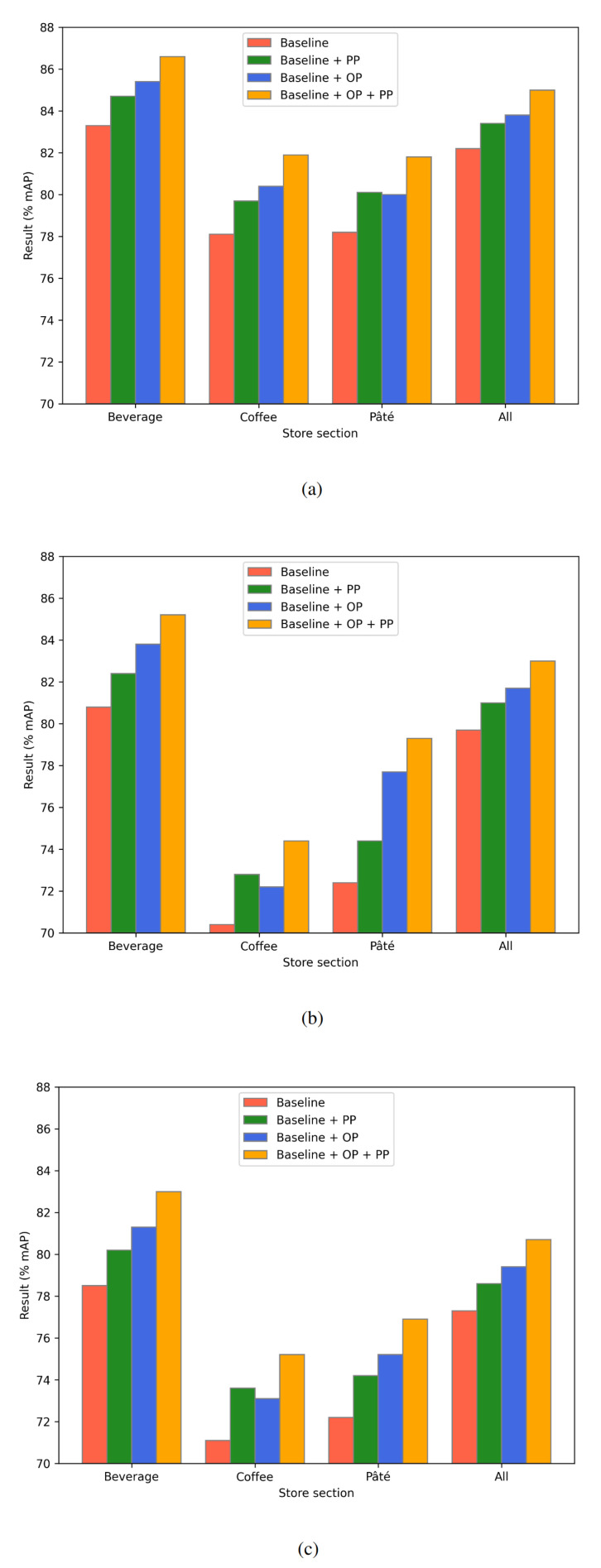
Ablation study of the proposed method for (**a**) YOLOv5, (**b**) YOLOv7, and (**c**) EfficientDet models. Each result represents the average mAP percentage of the five test folds. OP and PP represent the optimally pre-trained model and the use of post-processing, respectively.

**Figure 7 sensors-24-00693-f007:**
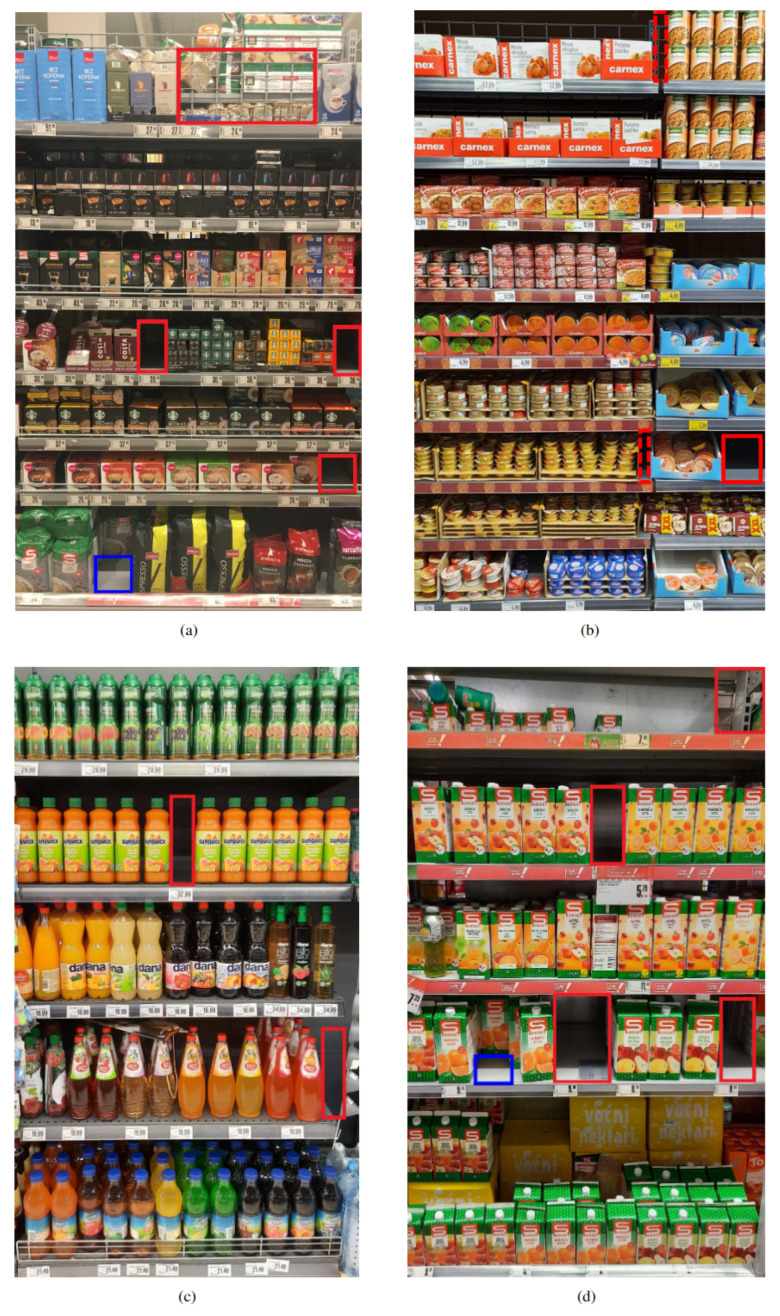
Examples of OOS detection results. The top row (**a**,**b**) displays the successfully analyzed images, whereas the bottom row (**c**,**d**) shows images with partially inaccurate results. The detected OOS instances of the normal and front classes are marked with red and blue bounding boxes, respectively. In (**b**), the dashed bounding boxes represent the OOS instances that were detected by the model but discarded after the post-processing was applied.

**Table 1 sensors-24-00693-t001:** Images and OOS classes distribution for each store section.

		Store Section			Total
		**Beverage**	**Coffee**	**Pâté**	
Images		171	178	162	511
Class	Normal	291	213	226	730
	Front	151	109	220	480
	Total	442	322	446	1210

**Table 2 sensors-24-00693-t002:** Aspect ratio intervals of OOS instances for each class and store section.

Store Section	Class	
	**Normal**	**Front**
Beverage	0.27 < *a* < 4.53	0.18 < *a* < 2.00
Coffee	0.28 < *a* < 3.66	0.18 < *a* < 1.75
Pâté	0.23 < *a* < 3.72	0.17 < *a* < 1.88
All	0.23 < *a* < 4.53	0.17 < *a* < 2.00

**Table 3 sensors-24-00693-t003:** Results of the proposed method and existing deep learning-based OOS-detection methods. Single-class and multi-class performances were measured using average precision (AP) and mean AP (mAP), respectively. Each result represents the average AP or mAP percentage of the five test folds. F denotes the use of the proposed front OOS class.

Method	Class		
	**Normal**	**Front**	**All**
Chen et al. [9] (Canny)	63.8	-	-
Chen et al. [9] (SVM)	56.9	-	-
Chen et al. [9] (Color histogram)	61.4	-	-
Jha et al. [12] (YOLOv5)	82.4	-	-
Jha et al. [12] (YOLOv7)	79.7	-	-
Jha et al. [12] (EfficientDet)	77.1	-	-
Jha et al. [12] (YOLOv5 + F)	83.3	81.1	82.2
Jha et al. [12] (YOLOv7 + F)	80.7	78.7	79.7
Jha et al. [12] (EfficientDet + F)	78.3	76.3	77.3
Ours (YOLOv5)	85.5	-	-
Ours (YOLOv7)	83.0	-	-
Ours (EfficientDet)	80.2	-	-
Ours (YOLOv5 + F)	86.3	83.7	85.0
Ours (YOLOv7 + F)	83.9	82.1	83.0
Ours (EfficientDet + F)	81.3	79.5	80.4

**Table 4 sensors-24-00693-t004:** Results of the two-stage training procedure. Each result represents the average mAP percentage of the five test folds. The result of the optimal pre-training strategy for each store section and model is bolded.

Model	Store Section	Baseline [12]	Pre-Training Strategy				
			**Fixed # of Pre-Training Epochs**				**Early Stopping**
			**10**	**30**	**50**	**100**	
YOLOv5	Beverage	83.3	-	84.5	84.2	**85.4**	85.2
	Coffee	78.1	-	79.6	**80.4**	80.1	79.9
	Pâté	78.2	-	79.4	**80.0**	78.4	77.6
	All	82.2	**83.8**	83.0	82.5	82.1	81.4
YOLOv7	Beverage	80.8	-	82.5	**83.8**	82.2	81.4
	Coffee	70.4	-	**72.2**	71.2	70.8	70.6
	Pâté	72.4	-	77.6	**77.7**	74.5	71.0
	All	79.7	80.8	**81.7**	80.1	79.0	77.8
EfficientDet	Beverage	78.5	-	80.9	**81.3**	80.6	79.7
	Coffee	71.1	-	72.2	**73.1**	71.0	70.5
	Pâté	72.2	-	**75.2**	74.9	72.5	70.9
	All	77.3	78.8	**79.4**	78.6	77.1	76.0

**Table 5 sensors-24-00693-t005:** Comparison of using different data augmentation techniques in the two-stage training process. Each result represents the average mAP percentage of the five test folds.

Data Augmentation	Model		
	**YOLOv5**	**YOLOv7**	**EfficientDet**
Rotation	81.5	78.9	76.7
Cutout	82.0	79.6	77.1
Contrast	82.4	80.0	77.6
OOS mirroring (ours)	83.8	81.7	79.4

## Data Availability

Restrictions apply to the availability of these data. Data was obtained from Cloudonia d.o.o. and are available from the authors with the permission of Cloudonia d.o.o.

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
