# Peer review of "Enhanced Out-of-Stock Detection in Retail Shelf Images Based on Deep Learning"

_sensors, 2024, doi:10.3390/s24020693_

Round 1

Reviewer 1 Report

Comments and Suggestions for Authors

This paper proposes two innovations: a two-stage training method with data augmentation and a post-processing technique. However, there are some issues that suggest a lack of innovation in this manuscript:

(1) The paper needs to compare its data augmentation strategy with other existing strategies to demonstrate its advantages.

(2) It needs to validate the two-stage training method using algorithms of different types, not just limited to the YOLO series.

(3) The post-processing technique seems to be similar to the cross-validation method used in traditional machine learning. In conclusion, it is necessary to supplement the description of the innovation.

Reviewer 2 Report

Comments and Suggestions for Authors

The current manuscript has some novelty in proposed contribution. The experimental results provide fair comparison. It needs revision in terms of technical details before acceptance. Some comments are suggested.

1. The architecture of the used Deep network should be shown in figure format.

2. Describe the hyper-parameter optimization of the used deep neural network.

3. Discuss about the runtime of your proposed method briefly. Because it may be used in real application.

4. How did you select the threshold in the equation 1?

5. Your proposed approach can be used widely in image retrieval systems as preprocess. For example, I find two paper titled “Content based image retrieval based on weighted fusion of texture and color features derived from modified local binary patterns and local neighborhood difference patterns” and titled “Innovative local texture descriptor in joint of human-based color features for content-based image retrieval”, which has enough relation. Cite these papers and discuss about potential applications and future works.

6. Is your proposed approach sensitive to the input image size of color space? In other words, is the proposed deep model accepting images in each size and color space? Discuss briefly.  

Reviewer 3 Report

Comments and Suggestions for Authors

The paper is devoted to detecting cases of missing goods on the shelves of a shop by means of image processing with a deep learning neural network. All necessary parts are present, the explanations are complete, logic and language are good. The paper can be published.

Reviewer 4 Report

Comments and Suggestions for Authors

1. The concept of Deep Learning is not discussed here in the paper, it is better to explain how the input input images are processed in DL.

2. References of 2023 are fewer, as there has been a lot of research in DL. 

3. The results need to be represented graphically, this will attract the reader in a better way.

4. 

Comments on the Quality of English Language

English is fine with minor editorial formatting needed. 

Round 2

Reviewer 1 Report

Comments and Suggestions for Authors

The manuscript is well revised, and it is acceptable in its current form.